# The Influence of Biologically Active Substances Secreted by the Adipose Tissue on Endometrial Cancer

**DOI:** 10.3390/diagnostics11030494

**Published:** 2021-03-11

**Authors:** Kaja Michalczyk, Natalia Niklas, Małgorzata Rychlicka, Aneta Cymbaluk-Płoska

**Affiliations:** Department of Gynecological Surgery and Gynecological Oncology of Adults and Adolescents, Pomeranian Medical University, Al. Powstańców Wielkopolskich 72, 70-111 Szczecin, Poland; nniklas488@gmail.com (N.N.); mal_ryc71@wp.pl (M.R.); anetac@data.pl (A.C.-P.)

**Keywords:** endometrial cancer, adipokines, cytokines, inflammation, angiogenesis, leptin, galectin, VEGF, FGF

## Abstract

Endometrial cancer is one of the most frequently diagnosed gynecological neoplasms in developed countries and its incidence is rising. Usually, it is diagnosed in the early stages of the disease and has a good prognosis; however, in later stages, the rate of recurrence reaches up to 60%. The discrepancy in relapse rates is due to the heterogeneity of the group related to the presence of prognostic factors affecting survival parameters. Increased body weight, diabetes, metabolic disturbances and estrogen imbalance are important factors for the pathogenesis of endometrial cancer. Even though prognostic factors such as histopathological grade, clinical stage, histological type and the presence of estrogen and progesterone receptors are well known in endometrial cancer, the search for novel prognostic biomarkers continues. Adipose tissue is an endocrine organ involved in metabolism, immune response and the production of biologically active substances participating in cell growth and differentiation, angiogenesis, apoptosis and carcinogenesis. In this manuscript, we review the impact of factors secreted by the adipose tissue involved in the regulation of glucose and lipid metabolism (leptin, adiponectin, omentin, vaspin, galectins) and factors responsible for homeostasis maintenance, inflammatory processes, angiogenesis and oxidative stress (IL-1β, 6, 8, TNFα, Vascular endothelial growth factor (VEGF), Fibroblast growth factors (FGFs)) in the diagnosis and prognosis of endometrial cancer.

## 1. Introduction

Endometrial cancer (EC) is one of the most common gynecological malignancies affecting women in developed countries. Its incidence rate is still rising and the number of new deaths is expected to increase by 17.4% by 2025 [1]. Endometrial cancer is a heterogenous disease and histological type accounts for much of the variation in responses to standard treatments. Bokhman classification is now considered too dualistic but, broadly, the division into endometrioid and non-endometrioid histological subtypes remains. However, within the endometrioid category, which accounts for the majority of endometrial cancers and carries the best prognosis, there are also some cancers that behave adversely. The newly introduced molecular subtyping into four categories (viz. in the exonuclease of domain of the polymerase epsilon gene “POLE ultramutated”, microsatellite instability hypermutated, copy number low and copy number high) is expected to better reflect the biological behavior of these cancers and, if performed on diagnostic specimens, could be integrated into staging and treatment pathways. Molecular analysis will supersede or complement the established risk factors of grade, lymphovascular invasion and depth of myometrial invasion that are currently used in addition to stage and histology in the planning of adjuvant treatment and post-treatment surveillance. Non-invasive biological markers have been investigated and some clinicians apply CA125 and HE4 as serum markers. Obesity is a major driver of the common variant, that is, endometrioid cancer, and biomarkers related to obesity and its associated proinflammatory state could be potentially used in endometrial cancer.

There are several mechanisms associated with obesity that could possibly explain the strong correlation between increased body mass and the prevalence of endometrial cancer. Adiposity increases the peripheral conversion of androgens and results in increased amounts of circulating bioavailable estrogens which are not counterbalanced by progesterone after menopause. Obesity-associated hyperinsulinemia may play an important role in endometrial cancer through a direct effect via the stimulation of endometrial cell proliferation or indirectly through sex steroid and insulin-like growth factor 1 pathways [2].

Adipose tissue acts as one of the major endocrine glands that synthesizes and secretes multiple hormones and cytokines that are involved in the regulation of glucose and lipid metabolism (omentin-1, vaspin, galectins, leptins, adiponectin), vascular homeostasis, immune regulation, apoptosis, angiogenesis and cell growth and differentiation (IL-1β, 6, 8, TNFα, Vascular endothelial growth factor (VEGF) and Fibroblast growth factor (FGFs)). In Table 1, we list factors secreted by the adipose tissue that take part in processes that may influence carcinogenesis due to their role in inflammatory processes, metabolism and angiogenic factors.

Obesity is associated with a chronic low-level inflammatory environment, which, together with potential paracrine factors secreted by the adipose tissue, could contribute to the process of carcinogenesis [3,4,5]. In this review, we sought to explore the impact of the factors in relation to the diagnosis and prognosis of endometrial cancer. For the purpose of our review, we conducted a literature search on PubMed, Web of Science, and Cochrane Library databases and included studies published up to November 2020. We evaluated the information provided in articles published in English using a combination of keywords relevant to the proper adipokines, angiogenic growth factors (VEGF, FGF and IGF-1), inflammatory cytokines (TNFα, IL-6, IL-1β and IL-8) and endometrial cancer. In this review, we focus on the factors secreted by the adipose tissue that seem to have the highest potential use in the diagnosis and prognosis of endometrial cancer.

## 2. Adipokines

Adipokines are polypeptide cytokines produced by the adipose tissue and they exert both systematic and local effects. Adipokines have been linked to cancer pathogenesis and progression as they impact on insulin resistance, inflammation and angiogenesis. Adipokines are especially important in obesity-related cancers as their levels were found to correlate with the amount of adipose tissue and patients’ BMI [6,7]. Unbalanced secretion of adipokines is associated with a spectrum of obesity-associated diseases. Most of the adipokines have pro-inflammatory properties and are increased in cancers, and some adipokines such as adiponectin, omentin and vaspin were found to be protective against malignancy. In Figure 1, we demonstrate the changes in adipokine levels associated with endometrial cancer.

### 2.1. Adipokines That Increase the Risk of Endometrial Cancer

#### 2.1.1. Leptin

Leptin is a peptide hormone secreted mainly by white adipose tissue and is primarily involved in energy homeostasis. It is encoded by the *ob* gene which is located on chromosome 7 [8]. It acts through transmembrane receptors (Ob-R) which belong to the class I cytokine receptor family. There are at least six receptor isoforms, with the main, fully active long form located in the hypothalamus, where it regulates the secretion of neuropeptides and neurotransmitters that balance appetite and body weight. Besides its crucial role in the central nervous system, leptin has peripheral effects, including its involvement in estrogen response or regulating insulin sensitivity [9].

Systematic levels of leptin correlate with the total amount of stored fat. Leptin acts as a pro-inflammatory cytokine contributing to the chronic inflammatory state in obesity. It shows structural similarities with interleukins (IL-6, IL-15, IL-12) and also with human growth hormone. It is generally considered that leptin has a crucial role in carcinogenesis, primarily by its ability to promote angiogenesis and influence many cellular pathways that could lead to cellular proliferation and suppression of cancer cell apoptosis [10,11,12].

Leptin receptor modulation promotes the development of endometrial cancer through the activation of JAK2/STAT3, MAPK/ERK, PI3K/AKT and COX-2 signaling pathways [13]. Higher serum leptin levels in patients with endometrial cancer are well documented [14], and may be considered as an independent risk factor for endometrial cancer [15]. Elevated leptin levels were also found in patients with pathological endometrial hypertrophy and atypical hyperplasia. A recent study showed the relationship between markedly elevated leptin levels in patients with higher staging of endometrial cancer (FIGO III and IV). Patients with higher serum leptin values had an increased risk for presence of lymph node metastases and infiltration of lymphatic vessels [16]. The overexpression of leptin receptors was documented in poorly differentiated endometrial cancer cells compared to healthy endometrial tissue samples. The depth of myometrial invasion correlated positively with leptin values and positive leptin and Ob receptor status was shown to greatly influence the 3-year survival rate of EC patients [17]. Zhoud et al. [18] supported the role of leptin in endometrial cancer progression by reporting that high Ob-R expression is inversely correlated with the degree of endometrial cancer histopathological differentiation. The role of leptin as a potential cell growth factor was reported in a study from 2013 [19] where it was proven that leptin, by increasing the expression of aromatase P450 and the local formation of estrogen, makes a significant contribution to enhancing the proliferation of endometrial cancer cells.

A meta-analysis by Ellis et al. [20] found that increased circulating levels of leptin and decreased levels of adiponectin are associated with an increased risk of endometrial cancer.

#### 2.1.2. Visfatin

Visfatin, also named nicotinamide phosphoribosyltransferase (NAMPT) due to its role in nicotinamide adenine dinucleotide (NAD) biosynthesis, is an adipokine secreted by the adipose tissue, although there is some evidence that it is mainly released by the visceral white adipose tissue macrophages in response to inflammation [21]. Many studies have reported the association between visfatin and systemic insulin resistance and hyperlipidemia [22] and increased circulating levels of visfatin in patients with obesity [23,24]. Visfatin was documented to be one of the most important risk factors for the occurrence of endometrial carcinoma [25], with its potential use as a prognostic factor and therapeutic target for EC [26]. Higher concentrations of vistafin were observed in subjects with blood vessel invasion and lymph node metastases in reference to the depth of infiltration of the endometrium and the tumor size. A high serum visfatin level could be used to prognose poor outcome in patients with endometrial carcinoma [27].

Visfatin is shown to significantly promote malignant progression of this cancer via activation of the insulin receptor (IR) and the phosphoinositide 3-kinase (PI3K)/AKT and mitogen-activated protein kinase (MAPK)/extracellular signal-regulated kinase (ERK) signaling pathways. Additionally, vistafin was proposed as a potential therapeutic agent in the treatment of endometrial carcinoma, as the pro-proliferative and anti-apoptotic effects of vistafin were abrogated by treatment with PI3K inhibitor and MEK inhibitor [28].

#### 2.1.3. Galectins

Galectins are carbohydrate-binding proteins that, through their carbohydrate recognition domains (CRDs), bind to beta-galactoside-containing glycans. To date, 11 galectins have been recognized in humans. Galectins are involved in i.a. the regulation of cell growth, differentiation, transformation, apoptosis and cell migration [29]. Due to their significant contribution to the induction of angiogenesis, tumor immune escape and metastasis formation, they are suggested to play an important role in carcinogenesis. Their diagnostic, prognostic and therapeutic value is still a subject of research.

A significant increase in the cytoplasmic staining of galectin-1 in endometrial cancer cells compared to normal adjacent endometrium was found by Van den Brule et al. [30] Further immunohistochemical analysis with biotinylated galectin-1 was also performed by Mylonas et al. [31]. Galectin-1 binding was demonstrated to be increased in FIGO stage III/IV, however, the results were insignificant. A correlation between galectin-1 binding and lymphangiogenesis was found, which could be used as a predictive factor of poor outcome in endometrial carcinoma. A recent study by Sun et al. [32] confirmed that galectin-1 levels are increased in patients with endometrial carcinoma and are associated with poor prognosis. Galectin-3 expression seems to be reduced in endometrial cancer compared to normal endometrium [30].

Brustmann et al. [33] reported that galectin-3 immunostaining was significantly increased, from normal endometrium, simple hyperplasia, complex hyperplasia and atypical hyperplasia to endometrial carcinomas. Enhanced galectin-3 expression was observed in carcinomas and the immunostaining of stromal cells was decreased in the latter. Moreover, in endometrioid carcinomas, galectin-3 expression increased with tumor grade. Other studies have also noted an increased serum level of galectin-3 in patients with endometrial cancer. Its concentration strictly correlated with the degree of tumor advancement [16]. A recent study [34] analyzed the clinical implications of galactin-3 in patients with endometrial cancer. Apart from the higher galectin-3 levels in EC patients compared to normal endometrium, significantly higher values of galectin-3 levels were found in patients with obesity, type 2 diabetes and increased levels of C-reactive protein and in nulliparous vs. multiparous women. Moreover, it has been noted that serum galectin-3 levels were strictly related to endometrial cancer stage and varied significantly between patients with FIGO III and IV vs. FIGO I and II. Higher concentrations were also connected with the presence of lymph node metastases and lymphovascular space invasion (LVSI). Patients with poorly differentiated G3 cancer had significantly higher median galectin-3 levels (IIIB, LVSI+, G2-G3, median 22.6 ng/mL) than patients with well-differentiated cancer (IA, LVSI-, G1, median 15.7 ng/mL).

Galectin-7 is primarily involved in epithelial homeostasis, cell adhesion and migration and it might show pro- or anti-apoptotic effects depending on the conditions. The role of galectin-7 in endometrial cancer is the subject of research. The latest study [35] suggests that elevated galectin-7 concentrations can promote the development of endometrial cancer via increased cell migration and decreased cell attachment. Its concentration increases with tumor grade and might have an effect on the metastasis formation. Galectin-7 was proposed as a novel biomarker in monitoring endometrial carcinoma progression.

The role of galectin-9 in the process of carcinogenesis is being investigated, due to its assumed ability in cancer cell apoptosis, its role in metastatic spread of many cancers [36] and as a prediction marker of poor outcome [37]. Brubel et al. [38] investigated the potential diagnostic role of galectin-9 in gynecological disorders. Elevated galectin-9 levels were found in patients with endometriosis, pelvic pain and infertility-associated benign gynecologic conditions. High levels of galectin-9 were noted in patients with endometrial cancer and, moreover, its expression differed significantly in accordance with histological patients’ histopathological grading, staging, the degree of myometrial invasion and presence of lymph node metastasis.

### 2.2. Adipokines Which Reduce the Risk of Endometrial Cancer

#### 2.2.1. Adiponectin

Adiponectin is a 30 kDa protein located on chromosome 3, mainly secreted by the adipose tissue, and takes part in the regulation of glucose and lipid metabolism. It is an anti-inflammatory and anti-proliferative cytokine and its levels are depleted in obese patients and causes an alternation of other adipokines, including TNFα [39]. Adiponectin can influence carcinogenesis either through a direct action on the tumor cells via adiponectin receptors and their expression on tumor cells or through insulin-sensitizing effects [40]. A meta-analysis by Zeng et al. confirmed the clinical association between high levels of serum adiponectin and a reduced risk of endometrial cancer, especially in post-menopausal patients [41]. High serum adiponectin levels were found to be associated with decreased endometrial cancer risk [42]. The adiponectin/leptin ratio is considered as a marker of adipose tissue dysfunction which can elicit oxidative stress and systemic inflammation [42]. An elevated leptin/adiponectin ratio (L/A ratio) is associated with an increased risk of endometrial cancer in post-menopausal women and the L/A ratio seems to be a better predictor of estimated risk than either leptin or adiponectin alone [43]. Meta-analyses show that serum leptin, adiponectin and the leptin/adiponectin ratio could be used in post-menopausal women as diagnostic and prognostic factors in patients with EC [44,45,46].

In contrast to pro-inflammatory adipokines, the levels of circulating adiponectin are inversely correlated to patients’ BMI and insulin resistance. Obese patients were found to have lower plasma adiponectin levels as well as lower expression of adiponectin receptors AdipoR1 and AdipoR2. Adiponectin can inhibit tumor progression through the PI3K pathway as it signals through its receptors (AdipoR1/2), activating AMPK and inhibiting PI3K/AKT/mTOR signaling [40].

Decreased expression of tissue adiponectin receptors was found to be associated with a higher histological grading of endometrial cancer [47] as lower levels of AdipoR1 receptors were found in patients with greater myometrial invasion and lymph node metastasis [48]. A study by Chen et al. found an association between three adiponectin genetic variants ADIPOQ SNPs and reduced risk of endometrial cancer [49].

Decreased adiponectin and insulin resistance correlate with an increased release of other cytokines and adipokines, including TNFα and IL-6 [50].

#### 2.2.2. Omentin-1

Omentin is a novel adipocytokine produced by visceral fat and expressed in adipose tissue stromal vascular cells. It acts as an endocrine factor modulating systemic metabolism as a secretory factor. It also acts locally in the adipose tissue in an autocrine and paracrine manner [51]. Omentin was demonstrated to have anti-inflammatory and anti-atherosclerotic effects via AMP-activated protein kinase/Akt/nuclear factor-κB/mitogen-activated protein kinase (ERK, JNK and p38) intracellular signaling pathways [52]. It was reported that subjects with impaired glucose regulation have decreased serum omentin-1 levels and that the depleted levels may contribute to the development of insulin resistance, type 2 diabetes, obesity and metabolic syndrome. Omentin-1 serum concentration is negatively correlated to BMI, insulin resistance index (HOMA-IR), fasting insulin, plasma glucose, leptin, TNFα and IL-6 [53,54,55,56]. The role of omentin in carcinogenesis is yet not well understood. Omentin-1 has an important anti-inflammatory role in obesity, probably by increasing in Th-2 cytokines comprising IL-13 and IL-14. It is believed that levels of inflammatory cytokines are decreased when there is a high concentration of omentin-1 [57].

#### 2.2.3. Vaspin

Visceral adipose tissue-derived serine protease inhibitor (vaspin) is another adipocytokine, which in adult humans is expressed by both visceral and subcutaneous adipose tissue. It is an insulin-sensitizing and anti-inflammatory protein. On the cellular level, vaspin exerts an anti-inflammatory effect on vascular endothelial cells and smooth muscle cells. Elevated vaspin concentrations were noted in patients with impaired insulin sensitivity and obesity [58]. It was suggested that vaspin is involved in the function of the female reproductive system as it was found to be constitutively expressed by, e.g., the hypothalamus and ovaries [58,59]. Its expression in the ovaries varied depending on the day of the menstrual cycle and was found to be regulated by gonadotropins, IGF-1 and steroid hormones [60]. Lower levels of circulating vaspin levels are correlated with increased risk for endometrial cancer [61]. A study by Erdogan et al. revealed a correlation between low serum vaspin levels and an increase in endometrial cancer risk, independent of risk factors such as age, BMI, HOMA-IR and Quantitative insulin sensitivity check index QUICKI [61]. Cymbaluk-Płoska et al. [16] demonstrated a correlation between low serum vaspin concentration and higher endometrial cancer staging. Moreover, vaspin was found to have the greatest specificity and sensitivity for endometrial cancer (83% and 89%, respectively) when compared to other adipokines, including leptin, galectin-3 and omentin-1. It seems that low vaspin concentrations, similarly to adiponectin, may be protective of endometrial cancer. This may be due to a decreased role of obesity-associated hormonal pathways involved in carcinogenesis, such as hyperestrogenism, hypoestrogenism, hyperinsulinemia and hyperleptinemia, which are all altered by the function of adipose tissue.

Below, we present Table 2, which shows the summary of the direct and indirect roles of selected adipokines in the pathogenesis of endometrial cancer. Adipokines may act through multiple signaling pathways as well as induce an indirect effect on carcinogenesis and its progression through insulin resistance, inflammation and angiogenesis.

**Table 2 diagnostics-11-00494-t002:** Direct and indirect effects of adipokines on the pathogenesis of endometrial cancer.

Adipokines	Direct Effect
Leptin	predominantly through JAK/STAT pathway which modulates PI3K/AKT3 signaling [62,63]
Visfatin	promotion of cell growth via NF-κB/Notch1 [64]
Galectin	MAPK family signal transduction and cell proliferation [65]
Adiponectin	inhibits cell proliferation via ERK1/2-MAPK pathway [66]
Omentin	Stimulates apoptosis through the activation of JAK/STAT signaling pathway [67]
Vaspin	inhibits proliferation and chemokinesis through the inhibition of NF-κB/Notch1 pathway [68]

## 3. Adipose-Derived Inflammatory Factors

Obesity was found to be associated with chronic inflammation, which could be caused by increased levels of fatty acids and production of inflammatory cytokines as well as the influx of immune cells. Obesity was correlated with increased levels of inflammatory cytokines produced by the adipose tissue, including TNFα, IL-6, IL-1β and IL-8 [69,70]. As inflammation is a predisposing factor for carcinogenesis, tumor growth and metastasis, it is important to study the impact of obesity on the pathogenesis of endometrial cancer (see Figure 2). However, it must be noted that the high levels of inflammatory cytokines in obese people are not exclusively due to release by the adipose tissue cells. The major pathway that correlates with the cytokines and cancer formation is the NF-κB transcription factor, which is activated as a response to inflammatory molecules and growth factors. NF-κB was also found to be linked with cell proliferation and apoptosis as well as the formation of metastasis and angiogenesis [70].

### 3.1. Interleukin-1β

IL-1β is considered an acute-phase cytokine with tumorigenic properties and participates in immune and inflammatory processes. Overexpression of IL-1α and IL-1β, the agonists of IL-1, were demonstrated to promote tumor invasion and metastasis through the induction of the expression of growth factors [71]. Increased levels of IL-1β can promote angiogenesis and tumor progression [72]. An elevated expression of IL-1R associated kinase (IRAK1) was noticed in malignant EC tissues, when compared to normal endometrium. The levels of IRAK1 correlated with worse patient staging, tumor invasiveness, metastasis and poor prognosis [73]. A study by Gonzalez et al. [74] found that leptin stimulates the secretion of IL-1β, acting through its functional receptor expressed by endometrial stromal cells (ESCs) and endometrial epithelial cells (EECs) and upregulates the expression of IL-1R at the protein level.

### 3.2. Interleukin-6

IL-6 is a pleiotropic cytokine produced by monocytes, lymphocytes, fibroblasts and endothelial cells, adipocytes, inflammatory cells, liver and muscles [75]. Its increased levels were found to be associated with the adipose tissue inflammation and acquisition of the pro-tumorigenic phenotype of the adipocytes. IL-6 was found to promote carcinogenesis [69]. Qi et al. have demonstrated the role of IL-6 in the progression of endometrial cancer through the local biosynthesis of estrogen [76]. Moreover, it could directly stimulate the proliferation of EC cells and increase the autocrine feedback loop even once IL-6 was withdrawn from the medium [77]. The study found an implication of the ERK-NF- κB pathway as a critical mediator of IL-6 production and its potential role in targeted treatment for patients with endometrial cancer. A study by Bellone et al. evaluated serum IL-6 levels as well as IL-6 gene expression in tumor tissue of patients with uterine serous papillary carcinoma (USPC) and patients with endometrioid carcinoma. The researchers found a significantly higher concentration of IL-6 in patients with USPC and EC, when compared to a healthy control group [78]. Moreover, the mean serum level of IL-6 was 6.1-fold higher in USPC patients than in EC patients.

### 3.3. Interleukin-8

IL-8 a pro-inflammatory cytokine expressed by the macrophages and fibroblasts. It is a chemoattractant, which main role is to serve as a macrophage-derived mediator of angiogenesis [79] that was found to participate in tumor angiogenesis in many pathologies including breast, ovarian, cervical and endometrial cancer [80,81,82]. Bruun et al. have found that IL-8 is also produced and secreted by the adipocytes [83] and its levels increase with BMI and waist circumference [84] In a study on patients with endometrial cancer, Fujimoto et al. found a significant correlation between cancer staging and IL-8 expression. [85] Moreover, Kotowicz et al. [86] have found that elevated levels of IL-8 are associated with shorter disease-free and overall survival of patients with EC and elevated levels of IL-8 before treatment may serve as a poor independent prognostic factor for OS. Moreover, the study revealed a significant correlation between the IL-8 concentration and CA125 levels.

### 3.4. Tumor Necrosis Factor α (TNFα)

TNFα is an inflammatory cytokine secreted by the inflammatory cells and adipocytes as well as cancer cells. Increased levels of TNFα are associated with adipose tissue inflammation and inhibition of adipocyte differentiation. TNFα was found to have a role in cancer promotion and progression, including cellular transformation, proliferation, invasion and metastasis [69,87]. Despite having poor angiogenic activity itself, its angiogenic potential seems to be modulated via the induction of strong angiogenic factors such as IL-8, VEGF and basic fibroblast growth factor (bFGF) [88].

In studies concentrating on colorectal cancer, associations between TNFα and different adipokines were studied. It was demonstrated that TNFα levels correlated with leptin levels. Guadagni et al. [89] found that TNFα may be an independent predictor of increased leptin levels. A similar association may occur in patients with endometrial cancer, in which obesity is an important risk factor. Leptin and TNFα may act in synergy in the promotion of endometrial hyperplasia and carcinogenesis, as leptin promotes low-grade inflammatory processes and elevates TNFα levels [90]. A meta-analysis by Ellis et al. revealed no difference in cancer risk between patients with different TNFα and IL-6 levels. [20] There is limited research on the role of cytokines in the pathogenesis of endometrial cancer, as the analysis only included three prospective studies and one retrospective study. Research should be continued to demonstrate their role.

## 4. Angiogenic Factors Secreted by the Adipose Tissue

Angiogenesis is a critical process for tumor expansion and proliferation. The growth of adipose tissue is similar to the processes required for tumor growth, in which epithelial, as well as stromal cells, induce an angiogenic response [69]. Both adipocytes and inflammatory cells produce various angiogenic factors (including VEGF and FGFs), adipokines (leptin, resistin, adiponectin) and cytokines which stimulate angiogenesis and contribute to the creation of a pro-angiogenic microenvironment [91]. As angiogenic growth factors are highly expressed in endometrial cancer, anti-angiogenic targeted therapy may have potential in the treatment of patients suffering from endometrial cancer [92,93,94].

### 4.1. VEGF

The role of vascular endothelial growth factor (VEGF) in the promotion of new blood vessel formation (angiogenesis) and vascular permeability is well established. Ref. [95] Angiogenesis has important implications in the process of tumorigenesis as, through vascular network nutrients, oxygen and growth factors are delivered to the tumor tissues, promoting the metastatic spread of cancer. It is generally considered that the level of expression of angiogenic factors, among which VEGF is one of the most important, reflects the aggressiveness of the tumor [96]. Moreover, VEGF is able to influence the tumor microenvironment through its influence on the immune cells. Additionally, the VEGF receptor pathway activates a cascade of processes that promote endothelial cell growth and migration [97,98]. The activity of the soluble VEGF isoforms depends on the presence of Flt-1 or KDR/Flk-1 tyrosine kinase receptors of the endothelial cells [99].

Overexpression of VEGF is associated with poor outcome in patients with endometrial carcinoma [100]. A significant correlation was found between increased levels of VEGF and the clinical stage and histological grade of the tumor. VEGF overexpression was suggested to be an important marker for predicting disease-free 5-year survival rate in endometrial carcinoma [101]. Angiogenic VEGF is predictive of vascular and lymphatic invasion, the depth of myometrial invasion and lymph node metastasis [102].

The agents targeting VEGF molecules seem to have promising results in clinical trials. Bevacizumab, a recombinant humanized monoclonal antibody against VEGF-A, whose action is to inhibit endothelial and tumor cell activation and proliferation, has been subject to Phase II clinical trials in recurrent or persistent endometrial cancer. Despite conflicting results, the anti-angiogenic effect found in the treatment of recurrent endometrial carcinoma has shown encouraging outcomes in many studies and should be further evaluated.

### 4.2. Fibroblast Growth Factor

Fibroblast growth factors (FGFs) are a family of 22 signaling proteins which have diverse roles and functions in the human body. They can be classified based on the mechanism of their action as intracrine, paracrine and endocrine. FGF1, FGF10 and FGF21 are secreted by adipocytes and act as autocrine/paracrine adipokines [103]. FGF ligands bind to FGFRs, causing dimerization of the FGFRs and activation of FGFR kinase domains through transautophosphorylation and signal transduction using a variety of pathways [104]. Impaired functioning of FGF pathway signaling can cause increased cell survival, increased cell motility and tumor angiogenesis. There is increasing evidence that the FGF signaling pathway may have an important role in carcinogenesis, including endometrial carcinoma [105].

FGF1 and FGF2 appear to have the greatest roles in cancer as both of them can act as tumor growth factors and thus increase tumor growth and invasion [106]. Expression of FGF1 and FGF2 was found to correlate with grading, level of myometrial invasion and staging of patients with endometrial cancer [107,108]. Additionally, FGF21 was demonstrated to have potential use in the diagnostics of endometrial cancer, as its levels correlated with clinical staging and tumor grading [109]. Mutations of FGF receptors were noted in patients with endometrial cancer. FGFR2 mutations were found to be associated with shorter disease-free and overall survival of patients with early-stage endometrial cancer [110].

The FGF pathway also has an important role in angiogenesis. As demonstrated using in vitro studies, FGF can act in indirect synergism with vascular endothelial growth factor (VEGF), leading to a greater angiogenic response [100]. Integrated mechanisms of tumor angiogenesis using FGFR and VEGF receptor (VEGFR) suggest that upregulation of either FGF or its receptor may have a potential role in anti-VEGF therapy resistance [111]. As the FGF pathway plays an important role in tumor angiogenesis and angiogenic escape during the inhibition of the VEGF pathway, various FGFs should be investigated to establish their role in EC. As clinical results show that FGFR inhibitors such as brivanib, dovitinib and lenvatinib are active in EC, [104] the role of FGF and FGFR in EC and its treatment should be further investigated.

### 4.3. Insulin Growth Factor-1 (IGF-1)

IGF-1 is a mitogen that has an important role in cell regulation, proliferation, differentiation and apoptosis. An alteration in the IGFBP/IGF/IGF1R pathway creating membrane receptors and proteins that regulate the activity of growth factors was found to impact the induction of carcinogenesis [112]. The IGF system was demonstrated to be associated with obesity, diabetes and hyperinsulinemia, which are all factors associated with endometrial carcinoma. IGF can exert its mitogenic effect using endocrine, paracrine and autocrine mechanisms [112]. IGF-1 is mostly produced by hepatocytes in response to the activity of growth hormone (GH); however, it can be also synthesized de novo in various tissues including the ovary, endometrium, breast and lung, where it can participate in cellular growth and have an anabolic effect [113]. Crosstalk between sex hormones and the insulin/IGF axis has been identified, as estrogens were reported to increase the expression of IGF-1 in the uterus, while [114] progesterone was found to increase the synthesis of IGF-binding proteins and antagonize estrogen-induced cell proliferation [115,116]. To date, there is limited and inconsistent data on the role of IGF-1 in endometrial cancer [117,118,119,120,121]. The biological actions of IGF are mediated through its receptors, especially IGF-1R, which undergoes autophosphorylation of its domain and activates the RAS–RAF–MAP kinase and the PI3K–PDK1–Akt/PKB signaling pathways, which results in cell proliferation and a decrease in cell apoptosis [122]. A study by Joehlin-Price et al. found an association between BMI and EC IGF-1R expression, and higher IGF-1R expression was associated with better disease-free and overall survival of patients with endometrial carcinoma [123]. IGF-1R seems to be a potential target for therapies using specific monoclonal antibodies. In clinical trials, human monoclonal IGF-1R antibody was found to inhibit endometrial cancer proliferation [124]. Moreover, specific monoclonal antibodies inhibited IGF-induced proliferation in both types I and II of endometrial cancer [125,126].

In Figure 3 we demonstrate the potential interplay of pathways that link chronic inflammation and angiogenesis caused by substances secreted by the adipose tissue of obese patients with endometrial cancer.

## 5. Conclusions

In this manuscript, we reviewed the complex role of adipose tissue in the pathogenesis of endometrial cancer. The adipocytes secrete adipokines that regulate hormonal and metabolic homeostasis and have angiogenic and inflammatory properties. Multiple adipokines were found to correlate with endometrial cancer. They could potentially be used as markers in the diagnosis and treatment of EC patients. The most promising seems to be the leptin/adiponectin ratio, which allows the assessment of the impact of adipokines on various physiological functions. The creation of algorithms taking into consideration serum levels of multiple adipokines at the same time seems to be a valuable test for the diagnosis and prognosis of patients with endometrial cancer. In the Table 3 below, we summarize the discussed factors that could potentially serve in the future as diagnostic and/or prognostic markers for patients with endometrial cancer.

**Table 3 diagnostics-11-00494-t003:** Potential diagnostic and prognostic factors for patients with endometrial cancer.

	Proposed Factors	Explanation
Diagnostic factors	Adiponectin, Leptin, Leptin/Adiponectin Ratio Vaspin, Omentin	May serve as independent endometrial risk factors [51] Meet the criteria to be used as a good diagnostic test [20,75]
Prognostic factors	Visfatin, Resistin Galectin	Can possibly serve to predict patients’ staging [32,33] May have a prognostic value and aid in prediction of patients’ survival [127]

Despite the recent research that emphasizes the role of molecular biology in the differentiation between the types of endometrial neoplasms, Bokhman’s [128] division into types I and II of endometrial cancers is still in use. Based on the current literature, it seems that the correlation between overweight and obesity is certainly more significant for patients diagnosed with type I EC (commonly referred to as estrogen-dependent EC) and most researchers concentrate their studies on patients with type I EC. A study by Bjorge et al. [129] revealed that obesity also influences the risk of type II endometrial cancer, as obese patients were 1.94 times more likely to develop EC compared to patients with normal body weight. Similar results were noted by McCullough et al. [130]. Angiogenesis is often increased in the processes associated with malignancy formation. VEGF seems to be a promising marker of proliferative processes and assessment of the risk of atypical conditions of patients with endometrial carcinoma [131].

To date, there has been limited evidence of routine biochemical testing and the use of imagining studies in the diagnosis and follow-up of EC. It is important to develop non-invasive markers and strategies that will allow the selection of high-risk patients that need to undergo extensive surgery and adjuvant therapy, without exposing others (low-risk patients) to unnecessary harm. Some markers, already established in screening for other malignancies, have been used in the screening and diagnosis of endometrial cancer, including CA125 and HE4, which are commonly used in the diagnosis of ovarian cancer.

Serum biomarkers are simple and minimally invasive methods of patient sampling. There is an urgent need to find specific, low-cost and reproducibly measurable markers that will allow risk prediction, early detection, evaluation of prognosis and surveillance of patients with endometrial cancer. Adipokines and other factors such as VEGF and FGFs that are released by the adipose tissue seem to be promising targets for novel pharmacological treatment. VEGF and FGF are of high importance as they could be potentially used as molecular targets for endometrial cancer treatment.

Targeting hormonal imbalances and hyperactive proliferative pathways associated with obesity can significantly reduce the risk of endometrial cancer. The maintenance of a healthy body weight is a possible way to prevent EC. Dietary and lifestyle changes should be supported. Obese patients should be offered counseling and consider pharmacological/surgical treatment to reduce the impact of obesity on the formation of chronic diseases as well as cancer formation.

## Figures and Tables

**Figure 1 diagnostics-11-00494-f001:**
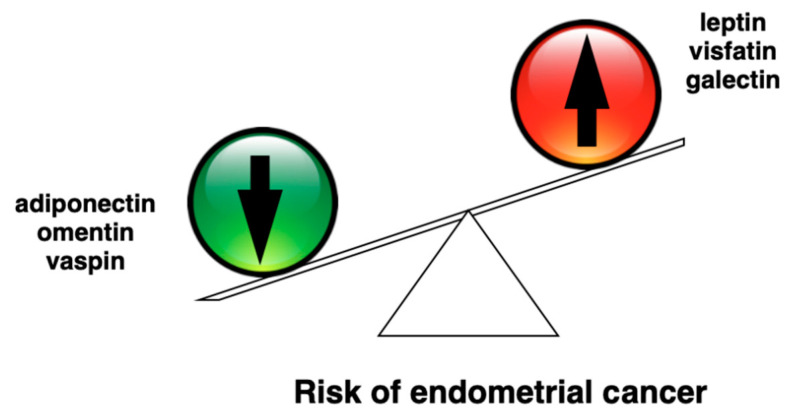
Changes in adipokine levels in patients with endometrial cancer (EC). The red arrows indicate elevated levels of certain adipokines in endometrial cancer, while green arrows determine the adipokines with decreased concentrations in the EC.

**Figure 2 diagnostics-11-00494-f002:**
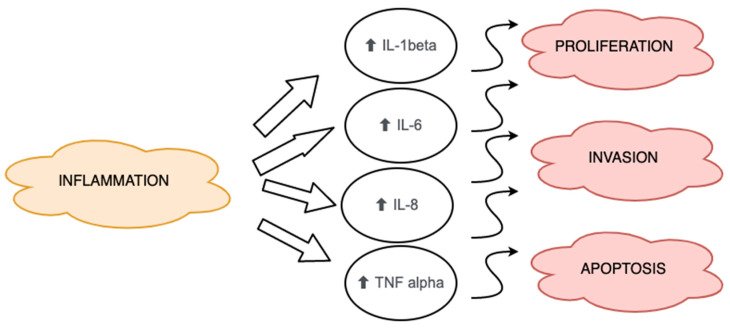
Changes in interleukin and TNFα levels due to chronic inflammation caused by obesity.

**Figure 3 diagnostics-11-00494-f003:**
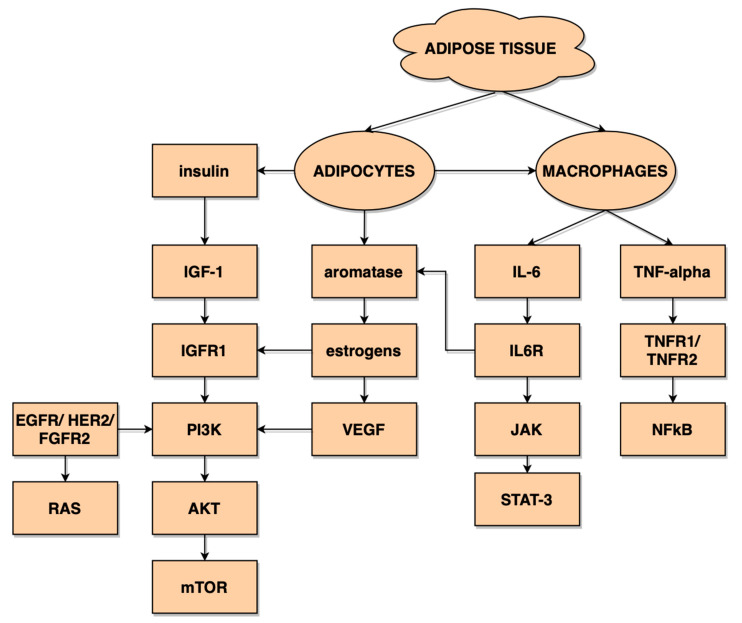
The interplay between the inflammatory and angiogenic factors secreted by the adipose tissue.

**Table 1 diagnostics-11-00494-t001:** Inflammatory, metabolic and angiogenic factors secreted by the adipose tissue.

**Adipokines**	Leptin Visfatin Galectin Adiponectin Vaspin Omentin Resistin Apelin
**Cytokines**	TNFα IL-1β IL-6 IL-8
**Angiogenic factors**	VEGF FGF IGF-1

VEGF: vascular endothelial growth factor, FGF: fibroblast growth factor, IGF-1: insulin-like growth factor 1.

## Data Availability

Not applicable.

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
