# Peer review of "The Influence of Biologically Active Substances Secreted by the Adipose Tissue on Endometrial Cancer"

_diagnostics, 2021, doi:10.3390/diagnostics11030494_

Round 1
Reviewer 1 Report
The authors reviewed the impact of Inflammatory, metabolic, and angiogenic factors, secreted by the adipose tissue, in diagnosis and prognosis of endometrial cancer.
Despite the clear commitment of the authors in writing this literature review, I do not believe that their work, as it stands, represents a significant improvement in knowledge in the field.
The manuscript, in fact, is not well organized and easy to follow in its path from data collection to conclusions.
In some parts, the discussion of individual factors focuses on their importance in tumors other than that of the endometrium and the reader loses contact with the meaning of the literature review.
In other parts the authors have misinterpreted what is written in the references cited.
The list of references is very poorly maintained with articles cited twice with different numbers and articles cited incompletely.
Specifically, some of my suggestions are:
- Focus the literature review on articles dealing with endometrial cancer and no other cancers;
- Reorganize the treatment of Adipokines by presenting first those that increase the risk of endometrial cancer (leptin, visfatin, galectins) and then those that reduce this risk (adiponectin, omentin, vaspin);
- If the authors want to refer to the nuclear factor kappa B (NF-kB) family of transcription factors (page 9, line 378) they should deal with the topic in more detail or not at all;
- In dealing with Interleukin 6 and Tumor necrosis factor α the authors should be more adherent to what was reported in the study by Bellone et al. (2005);
- The duplicate references are 34 and 43, 44 and 45, 47 and 48, 66 and 67, 98 and 99, 111 and 118;
- The references to be completed are 83 and 119.
Author Response
Dear reviewer, Thank you for your comments. We have improved the manuscript and hope it is now more suitable. Here are our responses to your suggestions:
- Focus the literature review on articles dealing with endometrial cancer and no other cancers; - Thank you for your suggestion; we deleted the parts talking about other cancers; we did that to show that these substances were not only analyzed for EC but also in other cancers and that may have some interest. However, we left the parts talking on how the adipokines influence glucose and fat metabolism as these are important factors for EC and the reason for the potential use of the adipokines as EC markers.
- Reorganize the treatment of Adipokines by presenting first those that increase the risk of endometrial cancer (leptin, visfatin, galectins) and then those that reduce this risk (adiponectin, omentin, vaspin);- the adipokines are listed in such way in the text as paragraph 2.1. talks about the ones increasing the risk of EC and paragraph 2.2. about these which are decreased in patients with EC. Moreover, we have changed the order of adipokines in Table 1. We’ve also changed the headings of the subparts.
- If the authors want to refer to the nuclear factor kappa B (NF-kB) family of transcription factors (page 9, line 378) they should deal with the topic in more detail or not at all; - thank you for your suggestion, we deleted the part
- In dealing with Interleukin 6 and Tumor necrosis factor α the authors should be more adherent to what was reported in the study by Bellone et al. (2005);
“ When IL-6 levels were quantified in the serum of endometrial cancer patients, we found significantly higher concentrations of IL-6 in both USPC and EC patients, when compared to the levels found in healthy control women or patients harboring benign gynecologic disease.” - this is the cite form the original article by Bellone et al. Moreover, we added additional information from the article stating that serum level of IL-6 was 6.1- fold higher in USPC patients than in EC patients.
- The duplicate references are 34 and 43, 44 and 45, 47 and 48, 66 and 67, 98 and 99, 111 and 118; -sorry, for the duplicated references, we improved that
- The references to be completed are 83 and 119. - we completed the references
Reviewer 2 Report
This is actually not a study, but a very thoroughly overview of factors involved in the regulation of glucose and lipid metabolism and factors responsible for angiogenesis, oxidative stress and inflammatory processes in adipose tissue.
It is interesting to look at the role of these factors in diagnosis and prognosis of endometrial cancer.
I have 2 main concerns with this paper (after editing the English language):
- I would prefer a systematic search on this subject in literature, most preferably a systematic review of the literature. How was the search for information done? Is it sure that all information available is included? Would it be possible to show the MSH terms etc. used for the search?
- It would be easier to read if there was a summary of factors involved in the diagnosis and factors involved in the prognosis separately of endometrioid endometrial cancer
Author Response
Dear reviewer, Thank you for your comments. We have improved the manuscript and hope it is now more suitable. Here are our responses to your suggestions:
It is interesting to look at the role of these factors in diagnosis and prognosis of endometrial cancer. - thank you for your comment; we submitted the manuscript as a review and not as an original research article
I have 2 main concerns with this paper (after editing the English language):
- I would prefer a systematic search on this subject in literature, most preferably a systematic review of the literature. How was the search for information done? Is it sure that all information available is included? Would it be possible to show the MSH terms etc. used for the search? -this was not a systematic review; we added to the main text information on how the research was conducted. “For the purpose of our review, we conducted a literature search on Pubmed, Web of Science, and Cochrane Library databases including studies up to November 2020. We evaluated the information provided in articles published in English using a combination of keywords relevant to the proper adipokines, angiogenic growth factors (VEGF, FGF and IGF-1), inflammatory cytokines (TNFα, IL-6, IL-1beta and IL-8) and endometrial cancer.”
- It would be easier to read if there was a summary of factors involved in the diagnosis and factors involved in the prognosis separately of endometrioid endometrial cancer- thank you for your suggestion; in the conclusions, we added a paragraph that shows the differences between factors associated with EC type I and EC type II.
Reviewer 3 Report
Endometrial cancer is the most common malignant neoplasm of female genital organs in developed countries. It is estimated that by 2025 the incidence of this cancer will increase by over 17%. Endometrial cancer is diagnosed in early stages in over 80% of patients. In these patients, the diagnostic methods and therapy are well known and the treatment results are very good. The challenge for gynecologists and oncologists are patients with advanced endometrial cancer and their aim is to search for new prognostic factors that could be a vector for new therapies in these patients.
The work submitted for review is a discussion of the current knowledge in this subject. The authors comprehensively presented the influence of factors secreted by adipose tissue including: leptyna, adiponectin, omentin, vaspin, galactins and related to homeostasis, inflammatory processes, angiogenesis and oxidative stress as diagnostic and prognostic factors in patients with endometrial cancer. In my opinion, in the near future they will also play a significant role in making therapeuticdecisions.
Author Response
Dear reviewer,
We would like to thank you for your comments. We improved the manuscript in accordance with the suggestions provided by other reviewers. We added an additional paragraph in the conclusions that highlights the differences of factors associated with the formation of EC type I and II.
Reviewer 4 Report
The Authors address a very important topic and this has the potential to be a very useful up to date summary the interplay of adipokines, cytokines and angiogenic factors in pathogenesis and prognosis of endometrial cancer. Dr Cymbaluk –Ploska has published substantial research papers.
With careful editing, attention to detail and language this has the potential to become a useful reference paper and should be considered for publication. It should not be published until it has been carefully edited by (I would suggest) the senior author.
In general it needs to be shorter by about one third and the number of references reduced by at least the same amount or halved. Some of the references are repetitive, some are numbered but absent eg #67, others have missing pages eg #109, or journal details eg#121, yet others are incorrect including some from their own department eg#26. This is too careless
This paper gives the overall impression that it has been submitted without appropriate senior supervision. I will edit some of the text to give a sense of how it can become more effective
The text needs to be rigorously edited by a senior author before re submission please
I wish the authors well with their resubmission
..........................................................................................................................................
Comment - I disagree strongly with lines 40-42
The Introduction needs to acknowledge that
Endometrial cancer is a heterogenous disease and histological type accounts for much of the variation in response to standard treatments. Bokhman classification is now considered too dualistic but broadly the division into endometrioid and non- endometrioid histological subtypes remains. However, within the endometrioid category which account for the majority of endometrial cancers and carries the best prognosis are some cancers that behave adversely. The newly introduced molecular subtyping into four categories (viz. POLE ultramutated, microsatellite instability hypermutated, copy number low, and copy number high) is expected to better reflect the biological behaviour of these cancers and if performed on diagnostic specimens could be integrated into staging and treatment pathways. Molecular analysis will supersede or complement the established risk factors of grade, lymphovascular invasion and depth of myometrial invasion that are currently used in addition to stage and histology in the planning of adjuvant treatment and post treatment surveillance. Non invasive biological markers have been investigated and some clinicians apply CA125 and HE4 as serum markers. Obesity is a major driver of the common variant, that is endometrioid cancer and biomarkers related to obesity and its associated proinflammatory state are candidate biomarkers.
Then follow with the later paragraphs from your Introduction
There are several biological mechanisms associated with obesity that could possibly explain the strong correlation between increased body mass and the prevalence of endometrial cancer. Adiposity increases the peripheral conversion of androgens and results in increased amounts of circulating bioavailable estrogens which are not counterbalanced by progesterones after menopause. Obesity-associated hyperinsulinemia may play an important role in endometrial cancer through a direct stimulation of endometrial cell proliferation or indirectly through sex steroid and insulin-like growth factor 1 pathways.14 Adipose tissue acts as one of the major endocrine glands that synthesises and secretes multiple hormones and cytokines that are involved in the regulation of glucose and lipid metabolism (omentin-1, vaspin, 81 galectins, leptins, adiponectin), vascular homeostasis, im mune regulation, apoptosis, angiogenesis and cell growth and differentiation (IL-1β, 6,8, 83 TNFα, VEGF and FGFs). In Table 1. we list factors secreted by adipose tissue that take part in processes that may influence carcinogenesis due to their role in inflammation, metabolism and angiogenic.
Obesity is associated with a chronic low-level inflammatory environ ment, which together with potential paracrine factors secreted by adipose tissue could contribute to the process of carcinogenesis.15–17 In this review we sought to explore the impact of the factors in relation to the diagnosis and prognosis of endometrial cancer. For the purpose of our review, we conducted a literature search on Pubmed, Web of Science, 94 and Cochrane Library databases including studies published up to November 2020. We 95 evaluated the information provided in articles published in English using a combination 96 of keywords relevant to the proper adipokines, angiogenic growth factors (VEGF, FGF 97 and IGF-1), inflammatory cytokines (TNFα, IL-6, IL-1beta and IL-8) and endometrial cancer. In this review, we focus on the factors secreted by the adipose tissue that seem to have the highest potential for use in the diagnosis and prognosis of endometrial cancer.
Adipokines
Adipokines are polypeptide cytokines produced by the adipose tissue and theyexert both systematic and local effects. Adipokines have been linked to cancer pathogenesis and progression as they imoact on insulin resistance, inflammation and angiogenesis. Adipokines are especially important in obesity-related cancers as their levels were found to correlate with the amount of adipose tissue and the patients’ BMI. Deranged secretion of adipokines is associated with a spectrum of obesity associated diseases. Most of the adipokines have proinflammatory properties and are increased in cancers, but some are such as adiponectin, omentin and vaspin have been found to be protective against malignancy. In Figure 1, we demonstrate the changes in adipokine levels associated with endometrial cancer.
2.1 Adipokines that increase the risk of endometrial cancer. 117
2.1.1. Leptin
Leptin is a peptide hormone secreted mainly by white adipose tissue and is primarily
involved in energy homeostasis. It is encoded by the ob gene which is located on chromo some 7.18 It acts through transmembrane receptors (Ob-R) which belong to the class I cytokine receptor family. There are at least six receptor isoforms with the main, fully active long form located in hypothalamus, where they regulate the secretion of neuropeptides and neurotransmitters that balance appetite and body weight. Besides its crucial role in the central nervous system, leptin has peripheral effects, including modulation of the estrogen response and insulin sensitivity.
Systemic levels of leptin correlate with the total amount of stored fat. Leptin acts as a proinflammatory cytokine contributing to the chronic inflammatory state in obesity. It shows structural similarities with interleukins (IL-6, 130 IL-15, IL-12) and also with human growth hormone. It is generally accepted that leptin has a crucial role in carcinogenesis through promotion of angiogenesis and other cellular pathways that lead to cellular proliferation and suppression of cancer cell apoptosis.20–22
The leptin receptor modulation promotes the development of endometrial cancer through activation of JAK2/STAT3, MAPK/ERK, PI3K/AKT and COX-2 signaling pathways.23 Higher serum leptin levels in patients with endometrial cancer are well-documented,24 and may be considered as an independent risk factor for EC.25 Elevated leptin levels were also found in patients with with pathological endometrial hypertrophy, which is a potential precancerous state. [Comment - What is pathological endometrial hypertrophy. Atypical hyperplasia is the premalignant condition]A recent study showed the relationship between markedly elevated leptin levels in patients with higher staging of endometrial 141 cancer (FIGO III and IV) and lower grading.
[This comparison of high stage and low grade is nonsense ]
Patients with higher serum leptin values had 142 an increased risk for presence of lymph node metastases and infiltration of lymphatic ves- 143 sels.26 The overexpression of leptin receptors was documented in poorly differentiated 144 endometrial cancer cells compared to the healthy endometrial tissue samples. The depth 145 of myometrial invasion correlated positively with leptin values and positive leptin and 146 Ob-Receptor status was shown to greatly influence the 3-year survival rate of EC patients. 147 27 Zhoud et al 28 supported the role of leptin in endometrial cancer progression by report- 148 ing that high Ob-R expression is inversely correlated with the degree of endometrial can- 149 cer histopathological differentiation. The role of leptin as a potential cell growth factor 150 was reported in the study from 2013 29 where it has been proven that leptin, by increasing 151 the expression of aromatase P450 and local formation of estrogen has a significant contri- 152 bution in enhancing the proliferation of endometrial cancer cells. 153 A metaanalysis by Ellis et al 30 found that increased circulating levels of leptin and 154
decreased levels of adiponectin are associated with an increased risk of endometrial can- 155 cer.
[This last paragraph makes no sense]
The whole text needs to be rigorously edited by a senior author before re submission please
Author Response
Dear reviewer, we would like to thank you for your comments.
The manuscript indeed needed some further editing. In accordance to your suggestions, we removed all of the unnecessary parts of the manuscript and hope it is now easier to follow.
Please see the improved version of the manuscript
We created the bibliography using Mendeley. We have some difficulties editing the bibliography and we would like to use the aid of the editorial team upon the acceptance of the manuscript
Reviewer 5 Report
The review is potentially interesting, the authors describe the principal inflammatory, metabolic and angiogenic factor secreted by adipose tissue and their role on endometrial cancer.
There are some deficiencies that can be improved.
- The first paragraph of section 2, line 281 is not sufficiently evidenced with references or results. It must be expanded and accompanied with more evidence. Either add to the conclusions or indicate this as a hypothesis.
- The high levels of inflammatory cytokines in obese people are not exclusively due to the release by the adipose tissue cells, this must be clarified, line 2402.
- Figure 3, line 2980, should be improved. It is not well understood that adipocytes and macrophages are cells of adipose tissue; as well as the arrow that joins them.
- The interleukin-1b and its role on endometrial cancer may be increased, section 3.
- The abbreviations used throughout the text are not well ordered. For example "EC" appears on line 43 and then "endometrial carcinoma (EC) appears on line 865. This should be modified.
Author Response
Dear reviewer,
We would like to thank you for your comments. We made the following changes to the manuscript:
- The first paragraph of section 2, line 281 is not sufficiently evidenced with references or results. It must be expanded and accompanied with more evidence. Either add to the conclusions or indicate this as a hypothesis. We added some references, however more references are provided in each specific paragraph regarding specific adipokines
- The high levels of inflammatory cytokines in obese people are not exclusively due to the release by the adipose tissue cells, this must be clarified, line 2402. - we added this information
- Figure 3, line 2980, should be improved. It is not well understood that adipocytes and macrophages are cells of adipose tissue; as well as the arrow that joins them. - we made changes to figure 3; please see the improved version of the manuscript
- The interleukin-1b and its role on endometrial cancer may be increased, section 3.
- The abbreviations used throughout the text are not well ordered. For example "EC" appears on line 43 and then "endometrial carcinoma (EC) appears on line 865. This should be modified. we corrected that, thank you for noticing
Reviewer 6 Report
In this review Kaja Michalczyk and co-workers summarize the role of active substances secreted by adipose tissue on endometrial cancer. They perform a list of adipokines, inflammatory factors and angiogenic factors and try to describe their role in EC development. Although the topic of this review is very interesting, in my opinion this manuscript is not exhaustive. Only one example: the role of VEGF in endometrial cancer have been analysed in more than 370 manuscripts but in this review it’s limited to a little paragraph. Some concepts have to be better described.
The review have several editing errors and English editing are necessary. Here some examples:
Line 301. Adipokines in which increase the risk of endometrial cancer.
Line 323. With with
Line 905 However, as enhanced galectin-3 expression was observed in carcinomas, immunostaining of stromal cells decreased in the latter
Line 1590-1591 Its levels are depleted in obese patients and cause an alternation of other adipokines including TNFα, risking their levels
Line 2404 metastasis…. Metastasis
……………………………………………………
Author Response
Dear reviewer, thank you for your comments
We improved the language used in the manuscript. Moreover, as suggested by other reviewers we made some additional changes to the manuscript. Please see the improved version.
In the review, we initially wanted to concentrate on the adipokines, however, other reviewers kept asking how other substances secreted by the adipose tissue influence endometrial cancer. We agree that the role of VEGF is of great importance and has the highest potential to be clinically used on a daily basis through the use of anti-VEGF drugs, however, in this article, we mainly wanted to concentrate on the adipokines and only mentioned other substances (cytokines and angiogenic agents) released by the adipose tissue
Round 2
Reviewer 1 Report
I thank the authors for the answers they gave to my suggestions and for the effort made to improve the comprehensibility of their work.
Regarding the answers to my suggestions 1, 2, and 3, I am satisfied with their answers and I believe that the revision of the text they made has greatly improved the manuscript.
Regarding the answers to my suggestion 4, in the paper from Bellone et al. “EC” means endometrioid carcinomas. Therefore, the sentence “A study by Bellone et al. evaluated serum IL-6 levels as well as IL-6 gene expression in tumor tissue of patients with USPC (uterine serous papillary carcinoma) and patients with endometrial carcinoma. (page 11 lines 477-479) should be rephrased as follows “A study by Bellone et al. evaluated serum IL-6 levels as well as IL-6 gene expression in tumor tissue of patients with USPC (uterine serous papillary carcinoma) and patients with endometrioid carcinoma.”
Regarding the answers to my suggestion 5, the duplicate references 66 and 67 are still there.
Finally, I suggest to take better care of references and describe references as reported in the “Instructions for Authors” for Journal Articles:
- Author 1, A.B.; Author 2, C.D. Title of the article. Abbreviated Journal Name Year, Volume, page range.
Note that page range is lacking in many references
Author Response
Dear reviewer thank you for your comments,
Regarding the answers to my suggestion 4, in the paper from Bellone et al. “EC” means endometrioid carcinomas. Therefore, the sentence “A study by Bellone et al. evaluated serum IL-6 levels as well as IL-6 gene expression in tumor tissue of patients with USPC (uterine serous papillary carcinoma) and patients with endometrial carcinoma. (page 11 lines 477-479) should be rephrased as follows “A study by Bellone et al. evaluated serum IL-6 levels as well as IL-6 gene expression in tumor tissue of patients with USPC (uterine serous papillary carcinoma) and patients with endometrioid carcinoma.” - Thank you for noticing; we changed “endometrial carcinoma” to endometrioid carcinoma”
Regarding the answers to my suggestion 5, the duplicate references 66 and 67 are still there. - we deleted the duplicated references, moreover we changed the formatting in accordance to the “Instructions for Authors” for the Journal. In case of any problems we will keep in touch with the editorial office.
Reviewer 2 Report
same comments as first version: no infornation on search for literature; other suggestion: to make at least a tables with infornation Diagnostics--factors-impact and Prognosis-factors-impact.
Author Response
Dear reviewer,
Thank you for your comments;
1. As we previously stated in the comments: this was not a systematic review; we added to the main text information on how the research was conducted. “For the purpose of our review, we conducted a literature search on Pubmed, Web of Science, and Cochrane Library databases including studies up to November 2020. We evaluated the information provided in articles published in English using a combination of keywords relevant to the proper adipokines, angiogenic growth factors (VEGF, FGF and IGF-1), inflammatory cytokines (TNFα, IL-6, IL-1beta and IL-8) and endometrial cancer.” We could additionally add the following information in the title- a literature review or underline that it was not a systematic review
2. we added a table including factors that could be potentially used either in the diagnosis of EC or serve as a prognostic factor. Please see the table (Table 3)
Reviewer 4 Report
Delete lines 29-41 in the Introduction
Can you reduce the Bibliography further please
Conclusion needs substantial work on language and text .
Author Response
Thank you for your comments.
We made some additional changes to the manuscript. Please see the reviewed version
Reviewer 6 Report
English and style editing are requested
Author Response
Thank you for your comments
Please see the improved version of the manuscript
Round 3
Reviewer 2 Report
none
Author Response
Thank you for your review